

# Measuring Decadal Vertical Land-level Changes from SRTM-C (2000) and TanDEM-X (~2015) in the South-Central Andes

Benjamin Purinton[1] and Bodo Bookhagen[1]

[1]Institute of Earth and Environmental Science, Universität Potsdam, Potsdam, Germany

**Correspondence:** Ben Purinton (purinton@uni-potsdam.de)

**Abstract.**

Vertical change is often measured in the cryosphere via digital elevation model (DEM) differencing to assess glacier and ice-sheet mass balances. This requires the signal of change to outweigh the noise associated with the datasets. On the ice-free earth, land-level change is much smaller in magnitude and thus requires more accurate DEMs for differencing and identification of

change. Previously, this has required high-resolution data at small scales. For the first time we measure land-level changes at the scale of entire mountain belts in the south-central Andes using the SRTM-C (collected in 2000) and the TanDEM-X (collected from 2010–2015), both spaceborne radar DEMs. Long-standing errors in the SRTM-C are corrected using the TanDEM-X as a control surface and applying cosine-fit co-registration to remove ~1/10 pixel (~3 m) shifts, Fast Fourier Transform and filtering to remove SRTM-C short- and long-wavelength stripes, and blocked shifting to remove remaining complex biases.

The datasets are then differenced and outlier pixels are identified as potential signal for the case of gravel-bed channels and hillslopes. We are able to identify signals of incision and aggradation (with magnitudes down to ~3 m in best case) in two > 100 km river reaches, with increased geomorphic activity downstream of knickpoints. Anthropogenic gravel excavation and piling is prominently measured, with magnitudes exceeding ±5 m (up to > 10 m for large piles). These values correspond to conservative rates of 0.2 to > 0.5 m/yr for vertical changes in gravel-bed rivers. For hillslopes, since we require stricter cutoffs

for noise, we are only able to identify one major landslide with a deposit volume of $16\pm0.15\times10^{6}$ m$^3$. Additional signals of change can be garnered from TanDEM-X auxiliary layers, however, these are more difficult to quantify. The methods presented can be extended to any region of the world with SRTM-C and TanDEM-X coverage where vertical land-level changes are of interest, with the caveat that remaining vertical uncertainties in primarily the SRTM-C limit detection in steep and complex topography.

## 1 Introduction

Geodynamic and geomorphological processes operating at different time-scales result in vertical change (herein $dh$) on the earth's surface. In the cryosphere, $dh$ studies use repeat surveys or digital elevation model (DEM) differencing on annual to sub-annual time-steps (e.g., Berthier et al., 2007; Nuimura et al., 2012; Neelmeijer et al., 2017; Brun et al., 2017). Changes to snow and ice occur most rapidly, but $dh$ measurement outside of the cryosphere provides aggradation and incision monitoring

for rivers (e.g., Mason and Mohrig, 2018), volumes of landslides and extruded lava (e.g., Bagnardi et al., 2016; Bessette-Kirton



et al., 2018), and earthquake displacements (Oskin et al., 2012). Large scale monitoring of $dh$ on soil, rock, and unconsolidated sediment is an elusive problem requiring signals that outweigh the noise in collection methods and resulting datasets.

Vertical accuracies for modern gridded spaceborne DEMs are on the order of 2–8 m in mountainous regions, though significantly worse on steepening slopes (e.g., Rexer and Hirt, 2014; Purinton and Bookhagen, 2017). Using DEMs from sources like the Advanced Spaceborne Thermal Emission and Reflection Radiometer (ASTER; Tachikawa et al. (2011)) with higher uncertainties is acceptable for monitoring glaciers and ice sheets (e.g., Brun et al., 2017), where $dh$ between even sub-annual time-steps can be tens to hundreds of meters over areas of many square kilometers. On the other hand, $dh$ of soil, rock, and unconsolidated sediment are often at the centimeter to meter scale and far more localized over up to a few hundred to thousand square meters. Due to these limitations, previous studies relied on intensive mapping from aerial photos (e.g., Hovius et al., 1997), sparse cross-sections with large temporal spans (e.g., Rinaldi and Simon, 1998), or—more recently—high-resolution topographic data from lidar or photogrammetric point clouds (e.g., Lane et al., 2003; Booth et al., 2009; Perroy et al., 2010; Cook, 2017) or very high-resolution optical satellites like Pleiades and WorldView (e.g., Bagnardi et al., 2016; Bessette-Kirton et al., 2018). Despite recent advances in lidar, high-resolution satellite, and unmanned aerial vehicle data availability (Passalacqua et al., 2015), these remain limited in spatial and temporal coverage, and sometimes prohibitively expensive. Coarser gridded DEMs from radar and optical spaceborne sensors remain the best, and often only, option in large or remote areas.

The publicly available Shuttle Radar Topography Mission (SRTM) DEM is an earth snapshot from its 10 day collection aboard the Endeavour Shuttle in February 2000. The mission produced an Interferometric Synthetic Aperture Radar (InSAR) DEM from C-band (5.6 cm wavelength) radar for 80% of earth's landmasses from typically 2–3 ascending and descending swaths (Farr et al., 2007). The SRTM-C has seen numerous succeeding releases and void filling (e.g., Jarvis et al., 2008). We use the most recent floating point re-processed 1 arcsec (~30 m) NASADEM, taking only the non-void filled original SRTM-C tiles (herein SRTM-C; Crippen et al. (2016); found in the "srtmOnly" directories under: https://e4ftl01.cr.usgs.gov/provisional/ MEaSUREs/NASADEM/). The TanDEM-X 0.4 arcsec (~12 m) DEM released in 2016—here received through scientific DLR proposals, though now available strictly commercially—is the next generation of radar-derived global topography following the SRTM. This DEM covering 97% of earth's landmasses was generated by semi-automated processing and stacking of > 470,000 ascending and descending X-band (3.1 cm wavelength) TerraSAR-X / TanDEM-X satellite bistatic scenes collected from December 2010 to January 2015 (Krieger et al., 2013; Rizzoli et al., 2017). As elevations are averaged between scenes, we take the date of the TanDEM-X as January 2015, thus providing a 15 year time step of $dh$ between SRTM-C and TanDEM-X. Using the latest possible date for TanDEM-X elevations means that rates of change are conservative minimum values.

In this submission we discuss the errors associated with each of these datasets and the corrections applied to mitigate uncertainties in their differencing for $dh$ detection outside of the cryosphere. We then difference the TanDEM-X and SRTM-C over the south-central Andes in northwestern Argentina (Fig. 1) to identify and measure areas of $dh$ in gravel-bed channels specifically and then across the landscape. Here, steep gradients in elevation (~1–4 km), rainfall (~0.1–1 m/yr), and vegetation (sub-tropical forests and croplands to arid, succulent-covered slopes) cause high rates of mass transfer (Bookhagen and Strecker, 2012; Savi et al., 2016; Schildgen et al., 2016), further influenced by climate change (Castino et al., 2016a, b, 2017)



and anthropogenic modification (gravel mining and weirs). To conclude, we discuss caveats driven by remaining uncertainties prevalent in spaceborne DEMs collected over complex topography.

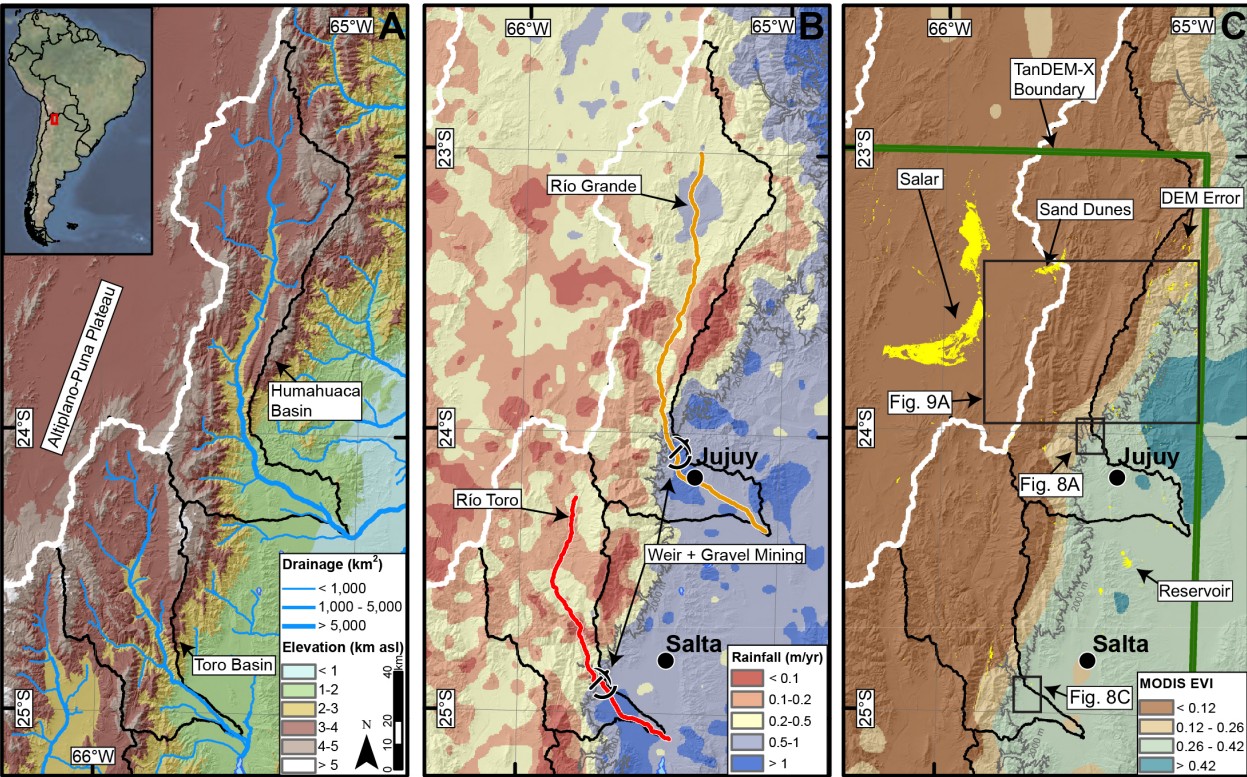

**Figure 1.** Overview of study area in NW Argentina with (A) elevation, (B) rainfall (Tropical Rainfall Measurement Mission 12 year average; TRMM2B31; Bookhagen and Strecker (2008)), and (C) vegetation (MODIS product 13C1 Enhanced Vegetation Index 14 year average; MODIS EVI; Huete et al. (1994)), where lower, brown (higher, green) values represent sparse (dense) vegetation. Note strong east-west gradients in all three maps. The white watershed boundary delineates the internally drained Altiplano-Puna Plateau. The gray line in (B) and (C) indicates the 2000 m contour line. The yellow patches in (C) are areas identified in the TanDEM-X water indication mask (WAM) as having low amplitude and/or low coherence. These patches correspond to salt flat (salar) regions on the plateau, water bodies (e.g., reservoirs in the low-elevation areas), steep and vegetated areas (DEM error), and other zones of coherence loss, such as the dunes identified. Inset boxes in (C) indicate locations of $dh$ map-view Figures 8–9, with TanDEM-X tile boundary in green. Note anthropogenic tampering of natural gravel-bed channels (Río Grande and Río Toro) with downstream flow diversion (weirs) and gravel mining activity nearby the populous cities of Salta and Jujuy.



## 2 Spaceborne DEM Errors

Yamazaki et al. (2017) classify spaceborne DEM errors into speckle noise, stripe noise, absolute bias, and tree height bias. We divide this further for the case of SRTM-C and TanDEM-X (both radar DEMs) into: (i) sensor specific related to radar and spacecraft collection, and (ii) terrain specific related to land-surface cover and topographic complexity. We do not consider

DEMs from optical sensors such as ASTER (Tachikawa et al., 2011) and the Advanced Land Observing Satellite (ALOS; Tadono et al. (2014)), which have well documented errors (e.g., Racoviteanu et al., 2007; Nuth and Kääb, 2011; Fisher et al., 2013; Yamazaki et al., 2017) and perform worse than radar, with vertical accuracies > 5 m (1-$\sigma$) and persistent high-frequency artifacts (Purinton and Bookhagen, 2017). On the other hand, within the study area, the SRTM-C and TanDEM-X both exhibit vertical uncertainties < 3.5 m (Purinton and Bookhagen, 2017) and also have an appropriately long time difference for

vertical land-level change detection. Auxiliary rasters including the water indication mask (WAM), height error mask (HEM), consistency mask (COM), and coverage map (COV) delivered with TanDEM-X (Wessel, 2016) allow enhanced understanding of DEM quality (cf. Supplement Section 1).

Random, or speckle, error caused by instrument thermal noise and localized de-correlation is the primary sensor bias for radar (Rodríguez et al., 2006). These localized, small magnitude errors reduce with increasing looks used in the final mosaic.

Speckle presents a greater issue in SRTM-C given the maximum three swaths at lower latitudes (Farr et al., 2007). Such noise is expected to be minimal in the TanDEM-X, with average coverage in our study area of seven ascending and descending scenes, and up to 14 in many steep areas (Fig. S1). Smoothing data prior to and after phase unwrapping (e.g., multi-looking, adaptive filters, or down-sampling) can further reduce speckle. The SRTM-C raw resolution of ~30 m is similar to the final 1 arcsec product, though, due to interferogram smoothing to reduce noise, the estimated true ground resolution of the final product is

45–60 m (Sun et al., 2003; Farr et al., 2007; Tachikawa et al., 2011). This may be improved in the newly released data (Crippen et al., 2016), but this remains to be tested. Multi-looking of 4×5 pixels of raw radar returns (resolution ~3.3 m) was used in the case of TanDEM-X to generate a final 0.4 arcsec (~12 m) product, thus significantly smoothing and reducing speckle (Rizzoli et al., 2017).

Besides a small geolocation error expected in both DEMs from instrument uncertainties, the SRTM-C has a number of

spacecraft specific biases, manifested in short- and long-wavelength striping (Rodríguez et al., 2006; Yamazaki et al., 2017). The short wavelength (~0.5–1 km, magnitudes typically < 0.5 m) stripes are related to jitter in the antenna mast caused by the periodic firing of shuttle attitude thrusters (Farr et al., 2007). Longer wavelength errors with magnitudes > 1 m are caused by individual swath tilts and form complex undulating patterns over ~100 km distances (Crippen et al., 2016; Yamazaki et al., 2017). TanDEM-X satellite biases can be found in slight tilting of individual TerraSAR-X / TanDEM-X scenes (e.g., Neelmeijer

et al., 2017), though these tilts were removed during stacking in the end product (Rizzoli et al., 2017). The careful monitoring and control maintained over flight geometry, in addition to post-processing to remove tilts using ICESat (Ice, Cloud and land Elevation Satellite; Schutz et al. (2005)), restricts most of the TanDEM-X uncertainty to the second category of terrain specific error (Rizzoli et al., 2017).



Land-surface cover plays a key role in modulating radar returns. TanDEM X-band and SRTM C-band radar have different penetration depths in dense vegetation (Carabajal and Harding, 2006; Hofton et al., 2006; Wessel et al., 2018) and snow and ice (Rignot et al., 2001; Rossi et al., 2016), leading to different height returns. We note this important caveat, but are able to ignore it for our particular study question (land-level change of bare material) and area (only partial vegetation and no permanent

snow and ice). Sub-tropical vegetation in our study area does allow some exploration of the effect on $dh$, however, we find no clear relation (cf. Supplement Section 2). In any case, vegetation differences are expected to be less significant than for optical data, which returns only the canopy heights (e.g., Yamazaki et al., 2017). Both DEMs have major inconsistencies and speckle over water bodies, wet salt flats, and deserts caused by de-correlation, variable reflectance, and/or weak backscatter of the radar signal (Rodríguez et al., 2006; Farr et al., 2007; Wendleder et al., 2013; Rizzoli et al., 2017). For the SRTM-C, these areas are

largely voids anyway, and for TanDEM-X the WAM raster provides information on coherence and amplitude for each pixel to identify these untrustworthy measurements (Fig. 1C).

Remaining errors in the SRTM-C and TanDEM-X are related to terrain characteristics (cf. Supplement Section 2). This is the result of topographic complexity below the resolution of the sensor, radar geometry considerations (layover, foreshortening, and shadowing), and interferometric phase unwrapping errors, all most pronounced in steep mountains. Such terrain biases are

demonstrated in the SRTM-C with elevation (Berthier et al., 2006; Paul, 2008), slope and aspect (Gorokhovich and Voustianiouk, 2006; Van Niel et al., 2008; Peduzzi et al., 2010; Shortridge and Messina, 2011), and resolution (manifested in curvature) (Gardelle et al., 2012), and in the TanDEM-X with only slope (Purinton and Bookhagen, 2017; Wessel et al., 2018). Terrain slope—also related to relief (Fig. S7)—is the primary cause of error in any DEM, demonstrated in the division of vertical uncertainties for most DEMs into slope bins (e.g., Wessel et al., 2018). Slope dependent errors may be reduced with higher

resolution data and increased look angles for mosaicking, as in the case of TanDEM-X, but these uncertainties are expected to remain as the most prevalent cause of error in any spaceborne DEM.

With this framework for understanding the potential error sources in the SRTM-C and TanDEM-X, it is possible to correct one dataset to another in a multi-step processing chain (e.g., Yamazaki et al., 2017) allowing $dh$ identification and measurement with greater certainty.

## 25  3   Correction of SRTM-C to TanDEM-X

Given the excellent agreement with differential GPS globally (Wessel et al., 2018) and in the study area (Purinton and Bookhagen, 2017) along with the minimal errors associated with orbital characteristics, we consider the TanDEM-X DEM as our reference surface in order to correct the more problematic SRTM-C. During correction, we do not apply any speckle reduction (e.g., via an adaptive filter as in Yamazaki et al. (2017)), as we are interested in raw elevation values and not a smoothed DEM.

For the SRTM-C we select the non-void filled NASADEM data so as not to include any auxiliary elevation measurements from, for instance, ASTER (Crippen et al., 2016). Importantly, both DEMs are referenced to the WGS84 ellipsoid vertical datum, whereas previous SRTM-C releases have been referenced to the EGM96 geoid (Farr et al., 2007), thus requiring a geoid-adjustment step introducing additional uncertainties prior to comparison.



For correction and differencing we use the 0.4 arcsec TanDEM-X that we bilinearly resampled to 1 arcsec to match the raw resolution of the SRTM-C. Wessel (2016) note that the delivered TanDEM-X 1 arcsec tiles, which we also have a number of, were generated with average resampling of the 0.4 arcsec tiles by DLR and not by any increase in multi-looks or interferogram smoothing. We tested a number of resampling schemes including average, bilinear, cubic, and cubic spline on the original

0.4 arcsec tiles and found better results (lower vertical uncertainty compared with differential GPS) from the commonly used bilinear resampling, whereas the un-edited 1 arcsec tiles delivered by the DLR—generated by average resampling—had higher vertical uncertainties.

The TanDEM-X and recently updated SRTM-C were both referenced to high-accuracy ICESat (Schutz et al., 2005; Zwally et al., 2009) measurements (collected between 2003–2009) during final block adjustments (Crippen et al., 2016; Rizzoli et al.,

2017). While this removes the complete independence of these datasets, the relative sparsity of these points (170 m along track and up to 80 km across track) does not provide a continuous adjustment surface, but rather acts to improve local elevations and overall DEM quality with respect to remaining tilts (Rizzoli et al., 2017). Throughout the study $dh$ refers to the TanDEM-X−SRTM-C 15 year differences (including both real change and vertical uncertainties).

## 4 Methods

### 4.1 SRTM-C Correction Steps

Our correction chain was applied using the previous SRTM-C output at each stage as input in the following step. All steps were carried out on a $1° \times 1°$ tile-by-tile basis (unprojected WGS84 vertical and horizontal datums), however, merging tiles and then processing produced identical results. We also found comparable results using Universal Transverse Mercator (UTM) equal area projected tiles. The correction steps served to correct SRTM-C orbital biases and did not attempt to correct for terrain

characteristics. We assumed that actual vertical change in our study area represented an extremely small fraction of pixels in the ~13 million pixel $dh$ raster for each tile. This ensures that the corrections only rectified SRTM-C biases on stable terrain and were not influenced by smaller areas of true vertical land-level changes. Comparison of correction steps was done using normalized percentage difference histograms and quantile-quantile (QQ) plots.

### 4.1.1 Co-registration

We corrected for sub-pixel offsets known to affect DEM comparisons (Van Niel et al., 2008; Berthier et al., 2007) using the universal co-registration of Nuth and Kääb (2011). This rigid translation is based on a cosine function fit to the relationship between terrain aspect and $dh$ normalized by terrain slope:

$$\frac{dh}{\tan(\alpha)} = a \cdot \cos(b - \psi) + c \tag{1}$$





where $\alpha$ is slope, $\psi$ is aspect, and the variables $a$, $b$, and $c$ are the magnitude, direction, and mean bias, respectively. The shifts were applied to the SRTM-C by bilinear resampling with the $dx = a \cdot \cos(b)$ and $dy = a \cdot \sin(b)$ vectors used to weight the neighboring cells, and the mean shift $dz = c \cdot \tan(\bar{\alpha})$ added at the end.

We fit equation (1) to only slopes $> 5°$ and, if necessary based on goodness of fit parameters, continued iteration of the fitting,
shift vector solving, and interpolation until the magnitude of the shift vector ($a$) was $< 0.5$ m or the reduction in normalized median absolute difference (NMAD; Höhle and Höhle (2009)) on stable terrain was $< 5\%$ (Nuth and Kääb, 2011).

Our co-registration did not correct for slope and curvature using polynomial fitting (e.g., Kääb, 2005; Gardelle et al., 2012) as this introduces empirical models and additional uncertainties. We did not note a linear positive or negative trend between slope and $dh$ (Fig. S7). Curvature versus $dh$ demonstrates the difference in actual resolution of raw sensor data between the
SRTM-C and TanDEM-X (Fig. S10), however, correction of this intrinsic measurement limit introduces artificial elevations and are thus inappropriate for $dh$ mapping between DEMs from different data sources and time-steps (cf. Supplement Section 2).

Iterative shifting and bilinear resampling of one DEM to another by decimeter steps had the same effect on rectifying aspect biases (same shift vectors leading to minimization of bias) as the empirical fitting of the cosine relationship and calculation
of shift vectors (cf. Supplementary Iterative Shifting Video). This indicates the robust nature of the method of Nuth and Kääb (2011), assuming a sufficient distribution of high-slope, multi-aspect-facing topography is available for cosine fitting. The minimization of the sum of errors and cross-correlation methods (e.g., Kääb, 2005) were unsuccessful at removing shifts in our study region.

### 4.1.2 Destriping

For removal of long- and short-wavelength striping patterns in the SRTM-C, we followed previous work using frequency analysis techniques to identify striping artifacts (e.g., Arrell et al., 2008) and noise (e.g., Purinton and Bookhagen, 2017) in DEMs. We took particular inspiration from Yamazaki et al. (2017) and used fast fourier transforms (FFTs) to filter the $dh$. In a first step, we removed all pixels identified as having low coherence in the TanDEM-X WAM. This filtered large water bodies and other areas that may show artifact noise affecting FFT analysis. Following this, any void pixels (including the low-
coherence areas) were set to $dh = 0$ and an FFT was run. The power spectral density (PSD) was calculated as the magnitude of the FFT squared and a mean $5 \times 5$ filter was passed over it. The ratio of original and smoothed PSD was then taken to identify regions of the spectrum with high outliers (high ratio) representing cyclic, tile-spanning stripe bias. We used the $97.5^{th}$ percentile of the ratio as the cutoff value. The remaining top 2.5% high- and low-frequency outliers received an inverse FFT, which produced a map of the long- and short-wavelength stripes. These stripes were then removed from the SRTM-C and the
process was repeated iteratively until the improvement in root mean squared error (RMSE) was $< 5\%$.

We refer to the above parameters as non-aggressive destriping, since we are just "shaving off" the top of the distribution. In aggressive tests, we experimented with lower percentile cutoff values (e.g., $95^{th}$) and lower tolerance for RMSE convergence (e.g., $< 2\%$ improvement). While these more aggressive destriping schemes did successfully eliminate the SRTM-C orbital biases, we also found that the true topography was often filtered following the $> 5$ iterations needed to meet the RMSE



convergence requirements (Fig. S11). Therefore, we chose to use the non-aggressive cutoffs and ran additional blocked shifting discussed in the following section.

### 4.1.3 Blocked Shifting

Patchy positive and negative regions in the co-registered, destriped $dh$ map were solved by breaking the $1° \times 1°$ tile into
square blocks and shifting each block by the median value. These areas likely correspond to remaining orbital biases that were not removed in our non-aggressive destriping technique. There may be local correspondence between these patches and atmospheric water vapor conditions at the time of SRTM-C collection in February 2000, however, such data at the fine scale necessary for analysis is unavailable. Furthermore, local adjustment of the SRTM-C and TanDEM-X to ICESat measurements could contribute to these shifts, though the contribution is difficult to quantify.

We began by masking the low-coherence pixels (again from the WAM) since these would disproportionately contribute to local median shifts. Using a variety of block sizes with edge lengths ranging from 1.35–7.2 km, we found the median $dh$ and median slope in each block. We used the median slope to normalize the median $dh$ values, since we expect areas of higher slope to have greater uncertainties and biases (Fig. S7) unrelated to SRTM-C orbital biases. Furthermore, we allowed a maximum shift per block of $\pm 1$ m, thus ensuring that this step did not cause unreasonably large shifts due to outliers contained in a given
block.

### 4.2 Differencing for Change Detection

Following orbital SRTM-C bias corrections, it is possible to merge corrected tiles and create maps of $dh$ to measure areas of actual change. Remaining uncertainties are primarily caused by terrain characteristics, with the biggest impact from slope.

### 4.2.1 Channels

We know from field observations that large braided gravel-bed channels in the study area (Fig. 1B) change rapidly with local incision and aggradation (natural and anthropogenic in the form of gravel mining) on the order of meters during the past decade. Trunk channel bank-to-bank outlines were hand-clicked from open-source satellite imagery from Bing™ and GoogleEarth™ in QGIS. We buffered the resulting channels by $-60$ m (upper limit of gridded SRTM-C resolution). This means we only use the wide ($> 120$ m), non-vegetated channel reaches from Río Toro and Río Grande where there has been recent aggradation
and incision.

Change mapping was done by separating the in-channel $dh$ values into bins of contributing error factors and applying $5^{th}$ and $95^{th}$ percentile cutoffs to each bin, thus only taking the top (positive=aggradation) and bottom (negative=incision) 5% of outliers. We first used the TanDEM-X WAM to remove the untrustworthy $dh$ pixels where coherence was lost three or more times (Wessel, 2016). Because gravel-bed channels represent a low-slope environment with no vegetation and we are
only measuring wide valleys, we assumed that DEM error from SRTM-C and TanDEM-X were restricted to random speckle noise. Nonetheless, to account for steeper areas with potentially more error from phase unwrapping, we separated $dh$ into




relief bins using the pixels' 500-m radius relief values. We also separated $dh$ by the TanDEM-X consistency (COM) and height error (HEM) masks (Fig. S2–S3). Taken together, $dh$ pixels in high-relief, high-height error, and low-consistency bins required greater magnitudes to avoid noise cutoffs than vice versa. A minimum level of detection (Lane et al., 2003) was taken as the RMSE of the entire $dh$ map on low-slope (similar to channel slope) areas. In a final step, all remaining in-channel $dh$ values

below this RMSE cutoff were removed as likely noise.

### 4.2.2   Entire Landscape

When considering $dh$ over the entire landscape, we include far more uncertainties related chiefly to steeper terrain. Thus, the error must be handled differently than for strictly low-slope pixels (in-channel). First, a corrected $dh$ map for the entire study area was generated. Similar to channel mapping, low-coherence pixels were removed with the WAM and $dh$ was separated

into bins of slope, height error, and consistency to retrieve only the top and bottom 5% of outliers in each bin set. The level of detection cutoff was taken as the RMSE across the entire landscape, which was almost entirely stable terrain, and remaining $dh$ values below this cutoff were eliminated.

At this stage, a great many lone and patchy $dh$ values remained. Given this, it was not possible to automatically identify areas of change that were only a small number of pixels in size. Interested in large-scale changes, likely not associated with a

single pixel, we sought connected pixels showing all up or all down vertical motion. To winnow the potential change pixels, we applied binary opening with a 1-pixel radius circular kernel, thus removing many unconnected outliers and small patches. Next, we took the summed $dh$ of each separate patch. It was assumed that the majority of patches, and thus majority of summed values, were remaining noise in the difference map, whereas signal should be spatially coherent and largely positive or negative. Therefore, by applying a standard deviation cutoff over summed patches (here we used 1-$\sigma$, though this can be easily set for

testing), we removed a vast majority of remaining pixels, and only kept the largest outliers. This limited the method to only assessing the largest coherent vertical changes in the landscape, but eliminated the possibility of mis-identifying change that was in fact noise. These remaining patches can be explored in map-view and compared with satellite or historical imagery for further confirmation and analysis.

## 5   Results

### 5.1   Correction Steps

Co-registration of SRTM-C to TanDEM-X revealed X-Y shifts of ~1/10 of a pixel (~3.7 m). Although minor Z shifts (~1 m) were also determined and corrected during co-registration, these were not unique across entire tiles, but rather related to long-wavelength SRTM-C biases. The cosine fitting to $dh$ normalized by terrain slope can be seen in Figure 2, whereas, in map-view the change is more subtle and difficult to discern.

In Figure 3, we demonstrate one iteration of destriping for a single SRTM-C tile (S 24°, W 66°). It is apparent in the co-registered $dh$ map that a number of long- and short-wavelength shifts are affecting the tile. Using our FFT, statistical cutoffs,



inverse transform, and stripe removal, the resulting $dh$ map has a much more uniform appearance and the median and RMSE are both reduced. This process was typically repeated 2–4 times per tile, until the RMSE began to converge. While topographic uncertainties remain in steep and high-relief regions, the overprinting biases are reduced.

Since we do not use an aggressive FFT filtering scheme, a number of patchy outliers remain. We attempted to correct these

regions using blocked shifting (Fig. 4), shown in this case over three tiles covering the foreland and Altiplano-Puna Plateau Region (S 24–26°, W 66°). After testing multiple block sizes, we preferred blocks with edge length of 3.6 km, since these provide a small enough area to correct highly localized inconsistencies, while also being far greater in size than the largest vertical changes we would expect in the landscape.

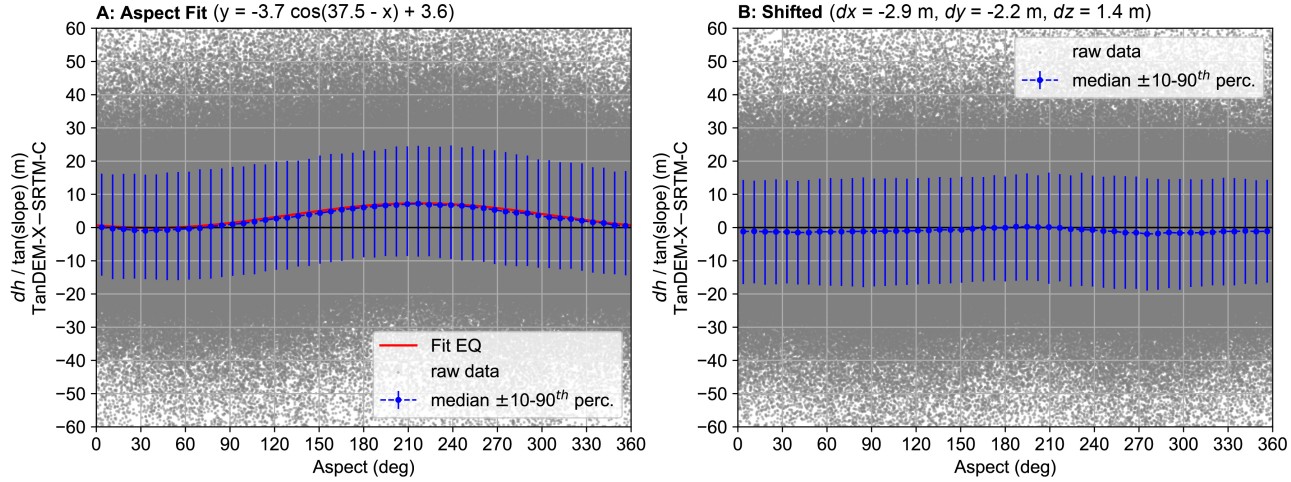

**Figure 2.** Relationship of $dh$ (normalized by tangent of slope) to aspect (A) before and (B) after co-registration and bilinear resampling of SRTM-C. We fit to equation (1) on all raw data. Note the close match between equation fit and median values. The cosine relationship in (A) is caused by overestimation of the SRTM-C on NE facing aspects (peaking at ~60°) and underestimation on SW facing aspects (peaking at ~220°). The resulting $(dx, dy)$ shift vector is directed SW.

### 5.1.1 Comparison of Correction Steps

Since stacked histograms are difficult to interpret and larger magnitude outliers are fewer in number and thus obscured, we plotted the normalized bin percentage difference of $dh$ in each step of correction (Fig. 5). Co-registration mostly caused a mean shift in the distribution. Moving to destriping, the number of pixels at high outlier values went down significantly (> 20% drop in ±15–20 m bins) and there was some (~10%) increase in bins ±5 m, whereas the number of values close to zero $dh$ decreased. This represents an overall re-distribution of error from the SRTM-C orbital biased patterns (Fig. 3) to a

more uniform spatial pattern (Fig. 4). The final blocked shifting caused very little overall change in the distribution, which was mostly in the form of another mean shift (this time directed the other way from co-registration). These effects can also





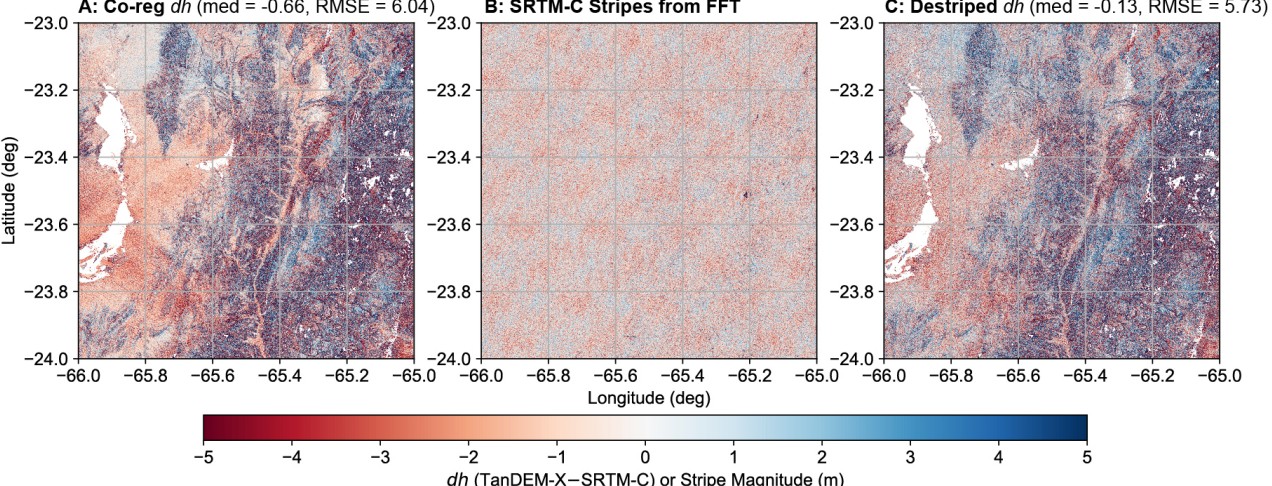

**Figure 3.** One iteration of FFT destriping from one tile (S 24°, W 66°). Both median and RMSE improve from (A) the co-registered map to (C) the destriped map. Stripes removed by FFT are shown in (B). Note that (C) is not the final corrected map as iteration was run twice more before RMSE began to converge at 5% tolerance level. Voids (white space) are untrustworthy pixels removed by TanDEM-X WAM cutoff prior to destriping.

be seen in a QQ plot of each subsequent correction step (Fig. 6), where co-registration caused a mean shift and some outlier reduction, de-striping had a large effect on narrowing the distribution at the tails, and blocked shifting again had a minimal effect on narrowing the distribution at the most extreme outliers. In all cases, the median value (0.5 quantile) moved closer to zero. Overall, these plots indicate the importance of SRTM-C correction and of the destriping step in particular prior to using

TanDEM-X−SRTM-C $dh$ maps for change mapping.

## 5.2    Areas of Change

As discussed in the methods, we separated potential change identification and measurement from corrected (co-registered, destriped, block shifted) $dh$ maps between the in-channel pixels and the entire landscape.

### 5.2.1    Channels

Binning corrected in-channel $dh$ and cutting off any remaining outliers within the low-slope RMSE of ~3 m reduced the data density significantly by cutting out any pixels within expected noise. The potential signal pixels were then plotted atop longitudinal profiles from the Río Toro and Río Grande (Fig. 7). The point clouds of $dh$ values were colored with a Gaussian kernel density estimate (KDE) to demonstrate the denser (warmer colors) versus sparser (cooler colors) zones of measurement. The density is displayed as percentiles of the full distribution of the 2D KDE of $dh$ from both channels. Turning to map-view,

we can observe the location of these pixels in the channel and their relation to local characteristics, upstream factors, and anthropogenic tampering (Fig. 8).



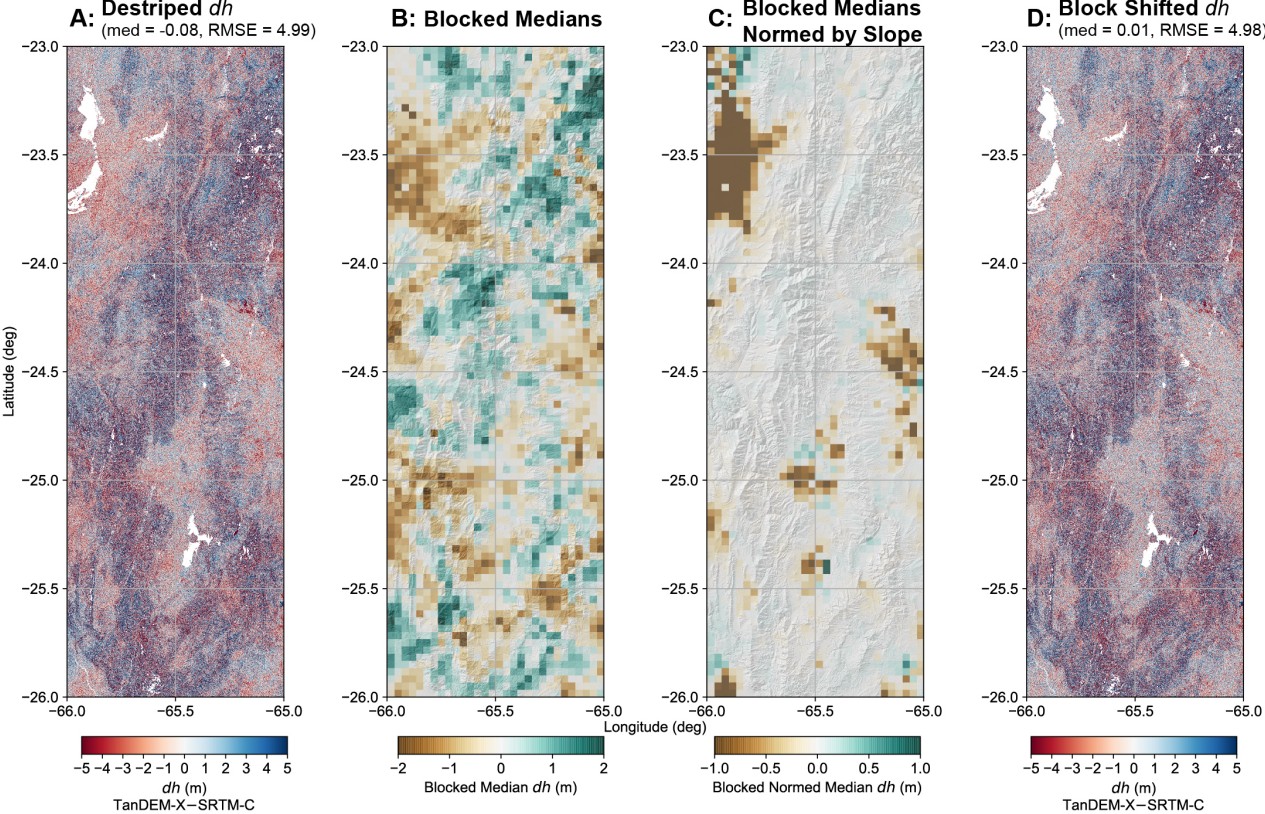

**Figure 4.** Blocked shifting on three destriped and merged tiles (S 24–26°, W 66°). Blocks are 3.6 km in height and width. The (A) destriped median and RMSE both improve slightly in (D) the final shifted $dh$ map. Note that the original blocked medians (B) show a slight pattern resembling the long-wavelength stripe bias from SRTM-C. In (C) we have normalized the median shifts by the median slope values, so as not to over-correct the steeper regions with higher uncertainties. The color scheme is changed for (B) and (C), and the scale of (C) is half the width of (B) since it only extends to the maximum allowable shift of ±1 m. Scales and color scheme in (A) and (D) are identical. Voids (white space) are untrustworthy pixels removed by TanDEM-X WAM cutoff prior to median calculation.

### 5.2.2 Entire Landscape

To be mapped as true vertical change, an area in the greater landscape must be significantly large and coherently positive or negative since many of the pure noise patches are > 10 pixels in size (> 0.01 km$^2$). Furthermore, the individual pixels must show significant height changes above the overall RMSE of ~6 m and outlier cutoffs in each bin, which in steeper bins may be > 10 m.

5 Examining results in map-view (Fig. 9) allows assessment of the potential true signal versus noise. At this stage it is necessary to include auxiliary data from field knowledge or remote sources like aerial or satellite imagery (e.g., GoogleEarth$^{TM}$). Our method was able to identify one major landslide (Fig. 9D), however, most other measurements are remaining large artifacts




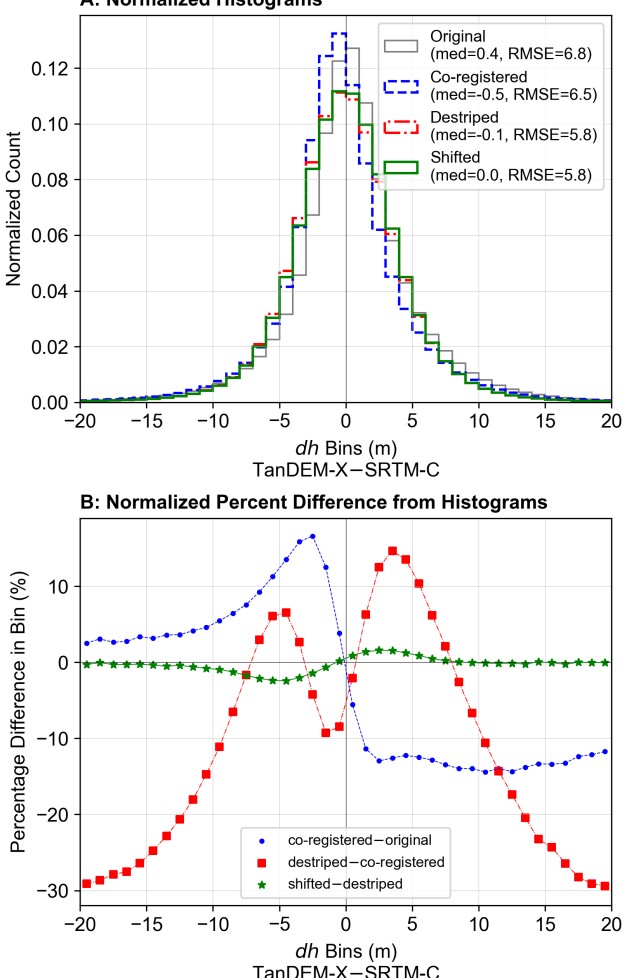

**Figure 5.** Stacked histograms (A) and normalized percentage bin difference (B). Though it is difficult to interpret the histograms, plotting their difference (normalized by bin count) as percentage change between successive steps demonstrates the shifting of the median to near-zero and the reduction in outliers.

attributable to both the SRTM-C and TanDEM-X. Low-coherence zones that may represent change between TerraSAR-X / TanDEM-X contributing scene collection (Fig. 9B–C) are necessarily removed in the WAM cutoff prior to binning.





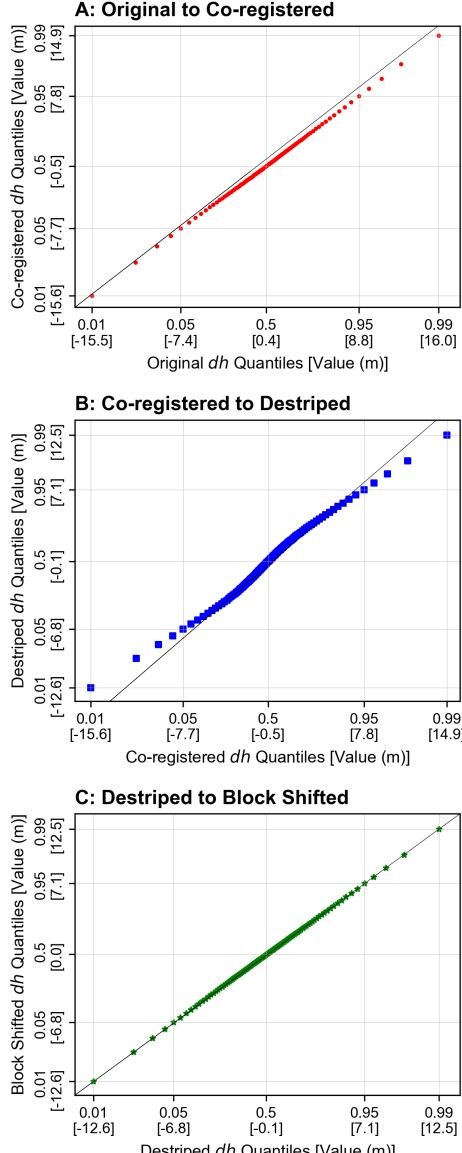

**Figure 6.** Quantile-quantile (QQ) plots showing difference between each successive correction step: (A) original to co-registered, (B) co-registered to destriped, and (C) destriped to block shifted. We note that co-registration and destriping have the greatest effect on zero-median-shifting and narrowing the outliers. The quantiles (0.01, 0.05, 0.5, 0.95, and 0.99) and their respective values are indicated on each axis to highlight this effect.





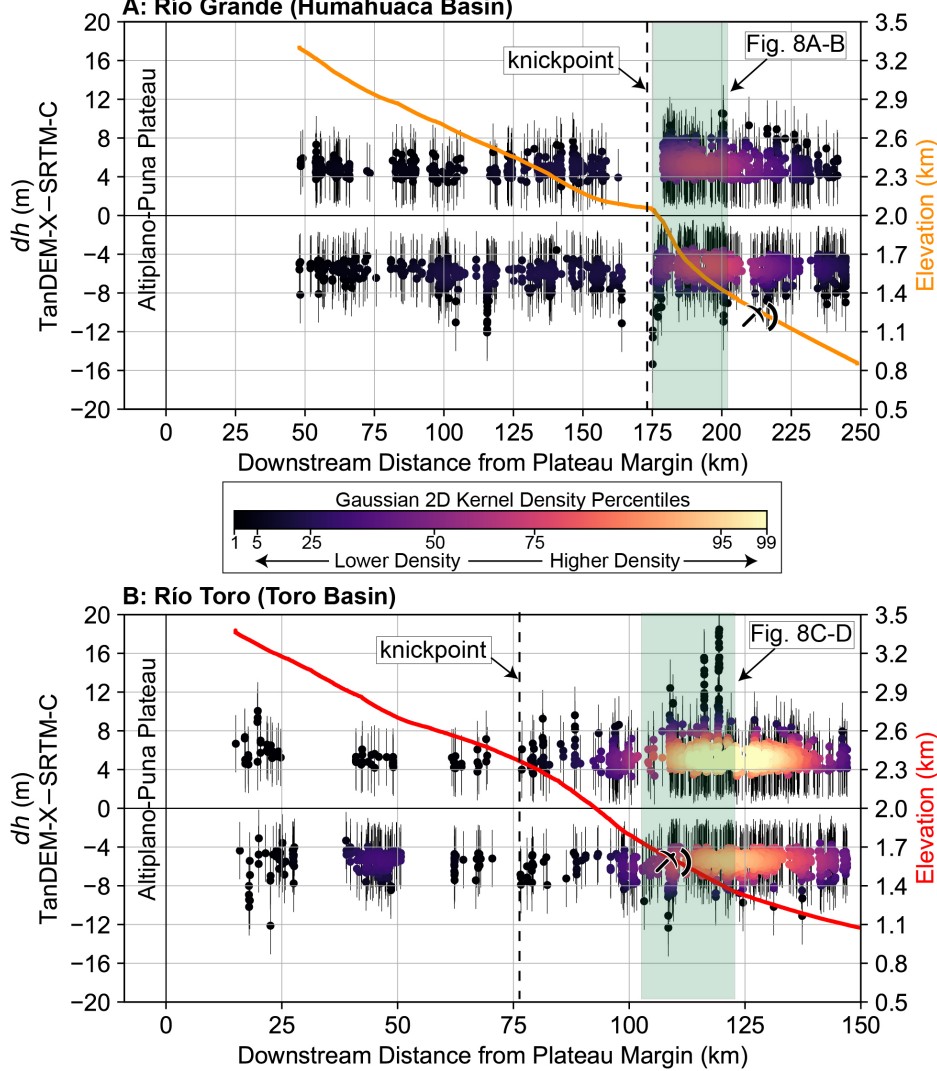

**Figure 7.** Longitudinal profiles of (A) Río Grande and (B) Río Toro overlain with point cloud of potential $dh$ signal (pixels outside of the range of expected noise). Error bars are RMSE from low-slope ($< 5°$) terrain outside of the channel area. Each $dh$ point cloud is colored by probability density from a Gaussian 2D KDE to show the denser (warmer) versus sparser (cooler) reaches. The KDE is scaled over all measurements from both channels and relative percentiles of the full distribution are used to highlight denser zones, particularly in (B) Río Toro. Note the $x$-axis range is 100 km greater for the longer Río Grande, despite the same axis scaling. Color scheme for elevation profiles on right axes match map-view color of each channel in Figure 1B. The knickpoint in Río Grande is caused by the large Del Medio fan (Savi et al., 2016), whereas the origin in Río Toro is tectonic, caused by the Gólgota Fault (Marrett et al., 1994; Hilley and Strecker, 2005). In both cases, the majority of the $dh$ signal appears downstream of the knickpoint. Map-view of green highlighted regions is shown in Figure 8.



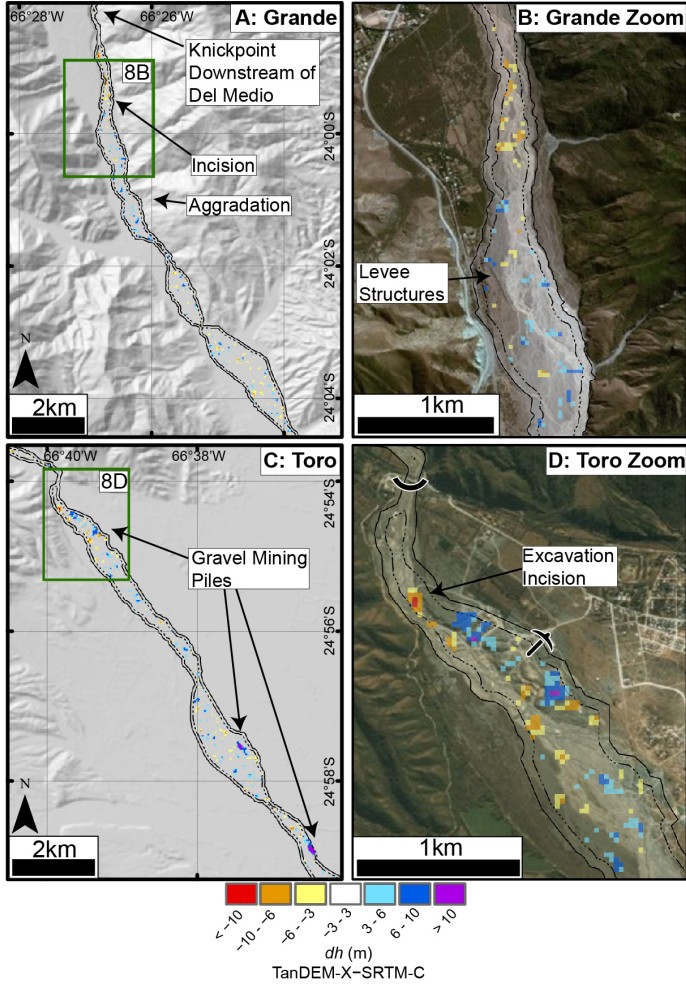

**Figure 8.** Map-views of the in-channel $dh$ measurements for Río Grande (A) and Río Toro (C) highlighted in the longitudinal profiles in Figure 7. For location of each map refer to Figure 1C. More details are shown in zoom-ins of the in-channel $dh$ measurements in (B) and (D). The solid outline is the hand-clicked bank-to-bank channel and the stippled line is the $-60$ m buffer area of measurement. We note large areas of incision related to the steep and narrow channel downstream of the Del Medio fan and knickpoint in Río Grande (A), immediately followed by a zone of aggradation with levee structures to direct gravels (B). For Río Toro (C) we highlight the anthropogenic influence of gravel mining generating large piles and also causing incision due to local excavation (D).

## 6   Discussion

### 6.1   Necessity of Correction Steps

The original SRTM-C is plagued by numerous terrain and sensor specific errors and biases (e.g., Carabajal and Harding, 2006; Gorokhovich and Voustianiouk, 2006; Van Niel et al., 2008; Gallant and Read, 2009; Yamazaki et al., 2017). Despite





**Figure 9.** (A) Map-view of landscape-wide $dh$ identification. For location refer to Figure 1C. Our method returns little change on the low-erosion Altiplano-Puna. The dunes (B–C) are not identified since they are masked out using the TanDEM-X auxiliary WAM as low-coherence zones. This indicates their rapid displacement between TerraSAR-X / TanDEM-X scene collection. Our method is able to identify one major landslide (D) in the Del Medio catchment (Savi et al., 2016), however, there are many erroneous results in steep and vegetated zones to the east, shown in (E) over the TanDEM-X hillshade.

re-processing of the original data in the new NASADEM product, many of these errors remain (Crippen et al., 2016). On the other hand, the newer TanDEM-X apparently has far fewer biases related to satellite geometry, and most error is restricted to terrain characteristics like slope and vegetation, though results are still nascent (e.g., Baade and Schmullius, 2016; Purinton and Bookhagen, 2017; Wessel et al., 2018). Our correction steps do not seek to eliminate bias related to terrain characteristics



at the scale of a few hundred meters, but rather to correct large scale biases related to primarily the SRTM-C at scales of several hundred meters to kilometers. Perhaps this reduction in bias is most obvious in map-view of the subsequent $dh$ patterns between processing steps (Fig. 3A to Fig. 4A to Fig. 4D), but we also show statistically that these steps lead to a narrowing of the distribution and centering of the differences on zero-median (Fig. 5–6). We assume that the vast majority of the pixels

(outside of the cryosphere) should be unchanged over 15 years, and thus median shifts between the datasets at large scales are biases in need of correction.

Co-registration indicates NE facing aspects are overestimated by the SRTM-C causing a negative excursion in the cosine fit, whereas SW facing aspects are underestimated and thus the $dh$ compared to TanDEM-X is positive. This error mostly affects higher slopes (Nuth and Kääb, 2011), which is the reason for normalization of $dh$ by the tangent of slope. The directions of bias

correspond to the look direction orthogonal to the SRTM-C descending path and parallel to the ascending path. This indicates that the source of this bias is the SRTM-C, as reported by previous authors (Bourgine and Baghdadi, 2005; Gorokhovich and Voustianiouk, 2006; Shortridge and Messina, 2011), and not TanDEM-X. A shift—accompanied by bilinear resampling—of just ~3.7 m (magnitude $a$ of equation (1) fit) to the SW rectifies this aspect bias.

As opposed to Yamazaki et al. (2017), we do not set a user defined ratio for FFT destriping, but rather use statistical

"shaving off" of only the outlier stripe noise until the data converges. This leaves some stripe noise at the expense of preserving topographic signal. In the case of more aggressive FFT filtering, using lower percentiles for the ratio cutoff and more strict RMSE convergence requirements, the actual topography began to filter out of the $dh$ maps (Fig. S11), which, as stated, is not the aim of our orbital bias correction steps and would lead to the inclusion of artificial (i.e., FFT generated) $dh$ measurements.

Remaining stripe noise is apparent in Figure 4B, where the blocked medians resemble the original long-wavelength stripe

pattern, though discontinuous. Despite the appearance in some areas of more negative values in the western parts of tiles (higher elevation, Altiplano-Puna Plateau), we do not find any clear relation between block medians and elevation at any block size or in any tile (cf. Supplement Section 3). Block shifting removes the remaining noise, but again we avoid correcting for strongly overprinting topographic biases related to slope by normalizing the block median $dh$ by median slope. Overall, these steps provide a more trustworthy $dh$ map, while respecting the inherent and difficult to account for biases in radar derived

spaceborne DEMs.

## 6.2  Potential Change Mapping

For lower slope regions (i.e., channels), the potential for change mapping is greater than in steeper areas. This is caused by the better agreement and lower vertical uncertainty of the two datasets in flatter, vegetation free areas. In both channels, the largest density of measurements is found below the respective knickpoints. This corresponds to an order of magnitude increase

in the 2D KDE shown by the warm colored patches in Figure 7. In terms of the actual number of measurements (number of $dh$ pixels) per binned channel reach, Figure S13 demonstrates this approximately five to ten fold increase in the downstream reaches with a simple histogram. This result partially has to do with a narrower channel and thus less measurements available above the knickpoints (hence the numerous gaps in measurement in the upstream reaches), however, these results also appear





to indicate that the most geomorphic work is happening downstream of the oversteepening point. This also coincides with a transition to a wetter environment in both cases.

The Río Toro has a particularly dense zone of measurements at the mountain front where naturally high rates of aggradation are enhanced by human gravel excavation and piling. On the other hand, in the Río Grande the downstream measurements

are spread over a greater channel reach and thus appear less dense in the 2D KDE (the measured Río Grande is ~100 km greater in length than the Río Toro). Downstream of the knickpoint, Río Toro is in a net aggradation state with a corrected $dh$ volume of $0.81{\pm}0.15{\times}10^6$ m$^3$, whereas, for Río Grande the net state is incision with a volume of $-0.69{\pm}0.15{\times}10^6$ m$^3$. In comparison, the pre-correction volume in each case is $-1.18{\pm}0.12{\times}10^6$ m$^3$ and $2.80{\pm}0.11{\times}10^6$ m$^3$ for Río Toro and Río Grande, respectively, thus indicating a flip in sign and halving of magnitude following careful corrections applied prior to

differencing.

Locally, the aggrading and incising patches may be related to braided channel avulsion and subsequent rapid incision into the unconsolidated bed material during frequent high-discharge events brought by convective rainfall in the summer monsoon (Castino et al., 2016a, b, 2017). In map-view (Fig. 8), we see that these automated measurements can be correlated with additional sources. For Río Grande, the steep knickpoint at the Del Medio fan (Savi et al., 2016; Schildgen et al., 2016) causes

a major zone of incision immediately followed by aggradation where the material is deposited. Fieldwork has indicated that some of this incision is man-made, caused by attempted removal of aggrading material coming from the productive (e.g., debris flows cf. Savi et al. (2016)) Del Medio catchment. Levee structures (Fig. 8B) are a testament to this tendency towards aggradation downstream of this extremely erosive fan. The cause of aggradation in the Río Toro is clearly enormous gravel piles being created just at and downstream of the mountain front. The volume of the large gravel pile indicated in Figure 8D

directly at the mountain front in Río Toro is $0.78{\pm}0.06{\times}10^6$ m$^3$, with this growth between SRTM-C and TanDEM-X observed during field work over the past decade and from GoogleEarth$^{\text{TM}}$ historical imagery back to 2003. This is coupled with incision in the active channel upstream of the piles where gravel is being removed to prevent widespread aggradation.

In terms of rates of change, our minimum measurable $dh$ of $\pm3$ m corresponds to a rate of $\pm0.2$ m/yr, given the conservative 15 year time difference between DEMs. The area of greatest point density in the longitudinal profiles in Figure 7 is centered

at $\pm5$ m, corresponding to a rate of $\pm0.33$ m/yr, with maximum rates of incision and aggradation, occurring at anthropogenic gravel piles and excavation sites, in excess of $\pm0.5$ m/yr. Human tampering is known to cause significant excursions from natural river dynamics (Kondolf, 1997; Grant, 2012), and we have shown that signals of excavation and piling are highlighted as above-the-noise outliers. Previous studies have demonstrated similar rates over longer time-scales (tens to hundreds of years) using more sparse measurements (e.g., Rinaldi and Simon, 1998; Rovira et al., 2005; Walter and Merritts, 2008; Comiti et al.,

2011) and at shorter time-scales (< 5 years) from high-resolution lidar data (Lane et al., 2003; Wheaton et al., 2010). The identification and quantification of incision and aggradation has important implications for infrastructure and agriculture given that 60% of global sediment delivery to coasts originates in high mountain regions (Syvitski et al., 2005).

Mapping $dh$ signals across the entire landscape presents a greater challenge given the higher uncertainties on steeper more complex topography. Nevertheless, using the binning method, binary operations, and outlier selection removes a large portion

of the noise from the corrected data. Our method displays very little change on the low-relief, low-slope Altiplano-Puna besides





some salt flat areas that were not removed by the coherence masking from the TanDEM-X WAM. Remaining noise mapped as potential change is clear at the mountain front where steep slopes and heavy vegetation causes complication of accurate radar measurement. In many locations these erroneous patches correspond with low-amplitude or low-coherence zones also identified in the WAM. We were able to automatically map one landslide, previously reported on by Savi et al. (2016), in the

Del Medio sub-catchment of the Humahuaca Basin using this method. This material likely contributes to the aggradation we see occurring downstream of the fan in the longitudinal profile (Fig. 7A) and in map-view (Fig. 8A). The calculated detachment and deposit volumes from this massive earth movement are $-10.5 \pm 0.12 \times 10^6$ m³ and $16 \pm 0.15 \times 10^6$ m³, respectively, with vertical land-level changes greater than $\pm 50$ m associated with the break-off and lobe (Fig. 9D).

    The area of sand dunes, clearly visible as a low-coherence region from the TanDEM-X WAM in Figure 1C and Figure

9B–C, is not mapped as potential change since the coherence masking prior to binning eliminates this area from consideration. Examination of $dh$ in this region is very noisy since the TanDEM-X contains measurements spanning 5 years, thus causing completely different height inputs for the same pixel in many scenes. This indicates the potential of the WAM alone for mapping change on shorter time-scales outside of very steep areas.

### 6.3   Caveats of Data and Method

Spaceborne DEMs present significant challenges for accurate height measurements, though until lidar or very high-resolution satellite data becomes more widespread and cheaper (Passalacqua et al., 2015), it is the only option in many study areas. On the other hand, unmanned aerial vehicles and point clouds generated using structure from motion technology could already provide a viable alternative (Javernick et al., 2014; Cook, 2017), but applying these methods at the scale of entire catchments or over tens-of-kilometers of river reaches is not feasible. Previously, $dh$ measurement from space has been primarily focused

on the cryosphere (e.g., Berthier et al., 2006; Nuth and Kääb, 2011; Neelmeijer et al., 2017) due to limitations in data accuracy. Certainly radar data are more adequate than optical data (e.g., Fisher et al., 2013; Purinton and Bookhagen, 2017) for the case of unconsolidated sediment, particularly since different penetration depths do not affect measurement (Rignot et al., 2001; Rossi et al., 2016), assuming limited vegetation.

    Here we have demonstrated the potential of new high-accuracy datasets such as TanDEM-X to correct outstanding biases

in the SRTM-C and potentially contribute to land-level change mapping and measurement over previously unattainable scales. Given remaining noise in the datasets, change mapping is limited to large areas of coherent change (e.g., massive landslides) or specific low-slope areas of interest such as wide gravel-bed rivers. In either case, field knowledge or auxiliary data (even in the form of GoogleEarth™) is necessary for accurate assessment of true change signals versus noise. In any case, the magnitude of change must be significantly above the expected uncertainty between DEMs, which in the case of SRTM-C and TanDEM-X

is as low as ~3 m on flat, partially vegetated terrain, and increasing with slope and topographic complexity. We posit that these correction steps may also be applied to cryospheric studies, though additional considerations for radar penetration depths must be accounted for.



## 7  Conclusions

In this study we have presented a novel use of two near-global spaceborne DEMs (SRTM-C and TanDEM-X) separated by ~15 years to measure land-level changes in the south-central Andes in northwestern Argentina. Previous measurement of land-level changes at the scale of entire mountain belts has been restricted to the cryosphere where the signal of snow and ice change outweighs the noise associated with DEMs used for differencing. On the other hand, outside of the cryosphere, studies have relied on higher resolution and higher accuracy data at much smaller scales to measure height changes in rivers and hillslopes. Using the TanDEM-X DEM as a control surface, we corrected long-standing SRTM-C errors related to orbital biases. We then successfully differenced the two datasets to identify and quantify land-level changes outside of expected noise caused by radar DEM speckle and other terrain dependent errors, increasing with steep and complex topography.

Our method is useful for the case of large gravel-bed rivers where the width far exceeds SRTM-C 1 arcsec resolution considerations. In such flat, vegetation free environments it is useful to analyze the river alone and not include additional uncertainties brought by increasing slopes. For these steeper regions, the use of greater outlier cutoffs and the necessity for large and coherent patches of land-level change, both to remove the majority of noise, limits the method to only very large earth movements. In either case, only signals outside of expected noise can be confidently identified, which in the case of gravel-bed rivers typically fall in the realm of human tampering. From the TanDEM-X auxiliary data alone it is also possible to identify regions that changed during TanDEM-X collection (2010–2015) using the water indication mask, however, this does not provide quantifiable change.

Overall, the use of relatively coarse (1 arcsec) spaceborne DEMs to derive land-level changes benefit from higher accuracy radar-derived data, whereas the use of optical data is limited to very high-resolution satellites. The application of this method to other regions around the world could indicate previously unmapped vertical changes. In the future, both the SRTM-C and TanDEM-X will continue to be used as snapshots of the earth's surface separated by over a decade, and thus useful for differencing against newer datasets yet to be developed to continue measuring vertical change outside of the cryosphere.

*Code and data availability.*  Python codes for co-registration, FFT destriping, blocked shifting, and potential change mapping are available on GitHub at https://github.com/UP-RS-ESP/SRTM-TanDEM-correction-dh.git. The SRTM-C updated NASADEM tiles can be found at: https://e4ftl01.cr.usgs.gov/provisional/MEaSUREs/NASADEM/. TanDEM-X data is only available from DLR commercially for the time being

*Competing interests.*  The authors declare no competing interests in the preparation or submission of this manuscript





*Acknowledgements.* The authors thank the DLR for TanDEM-X DEMs received through DEM_CALVAL1028 to B. Purinton and DEM_GEOL1762 to Stephanie Olen. Additional funding was sourced from DFG Graduate School StRATEGy (IGK2018) and NEXUS funded through the MWFK Brandenburg, Germany, both to B. Bookhagen.



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
