# Peer review of "Measuring Decadal Vertical Land-level Changes from SRTM-C (2000) and TanDEM-X (~2015) in the South-Central Andes"

_Earth Surface Dynamics, 2018_

## Referee Comment (RC1) · Anonymous Referee #1 · 14 Aug 2018

The study analyses elevation differences between the SRTM and TanDEM-X DEMs over mountain terrain. The processing procedure to reach maximum accuracy and to remove various horizontal and vertical shifts is sound, but the geophysical results are more sparse than expected. My comments:

(1) The study is mostly a method presentation, the results are less spectacular than the reader expects ("novel", "first time"). I strongly recommend to tone done the latter type of announcements to make the reader not expect dense measurements over an entire mountain region.

[Figure]

(2) Perhaps show and analyse (sediment fluxes) the entire river reach covered not only a subsection.

(3) Page5Line5, and else: land level changes are measured for instance using medium resolution ASTER data. Tome down "for the first time" if it refers to medium resolution. Also, are you sure that no landslides etc. have been measured using SRTM and TanDEM-X before? Also, TanDEM-X is in my view a high-resolution sensor, not a medium resolution one.

(4) P2L6: be less strict. ASTER timeseries detect a few cm/dm per year over 10-20 years (=dm-m total change)

(5) P2L14: what about ArcticDEM, HighMountain Asia DEM, ALOS PRISM AW3D? I wouldn't call them limited spatial coverage.

(6) Section 3 (Correction...) is rather part of the methods.

(7) Fig 8 and 9: you also need to show the results outside the masks in order to let the reader judge the statistical significance.

(8) Fig 8: I guess these magnitudes of changes would also be visible in ASTER time series, or ASTER versus SRTM (for instance;Castro et al. 2016 (Nature Comm Art 13585), Brun et al. 2017 (in your list), Girod et al. 2017 (doi: 10.3390/rs9070704), Wang et al. 2015 (doi: 10.3390/rs70810117))

(9) Fig 9: The landslide dh would also be visible without your processing, I guess. You could use this (and the river) to visualize the importance of your processing in more detail (before-after processing).

(10) P19L7-8: How are uncertainties computed? StDeviation, StError (if StError, how computed?). How aggregated and how voids filled?

(11) P30L31: it is not that easy to account for radar penetration! It exceeds often the actual elevation change signal, and the correction magnitudes applied here. See above

ASTER studies, and others.

Detail comments

page 1, line 2: rewrite 1st sentence. "Vertical change is measured in the cryosphere ...". I understand what you mean, but what is "measuring in the cryosphere"?

P1L4: typically much smaller (landslides, as the one you show later, are a frequent exception with vertical changes on the same order of magnitude as glaciers).

P18L15: leaves stripes noise at the expense of preserving ... ? Other way round?

---

## Referee Comment (RC2) · Anonymous Referee #2 · 26 Aug 2018

My primary expertise with respect to this manuscirpt lies in the technique, and subsequent interpretation, of geomorphic change detection. At the outset, I would therefore like to emphasise that my review focuses upon the overall form of the manuscript and the technical component of the DEMs of Difference analysis. I do not have the technical expertise to scrutinise the detail of the remote sensing data processing; other reviewers should be sought for this elements.

Overall, this manuscript presents an interesting and novel demonstration of how space-borne radar DEMs can be used to detect vertical change in the Earth's surface. How-

ever, in my opinion, for this journal the manuscript needs to be reorganised to present a clearer research question/aim at the outset that is focused upon the geomorphological problem that is being investigated. There are also elements of the context, methods and results that are not organised in a classical research paper order. For the material that is presented, I do not see a reason why the context, methods and results can't be split into separate sections. I elaborate on these two items below, in addition to identifying further major and minor points.

Major comments

1) A clear geomorphic research problem needs to be identified at the outset and backed up with appropriate context. P2L29 describes what will be included in the paper but there is a need for a more explicit geomorphic aim and associated set of objectives. The data processing methodology to generate a DEM of Difference is novel and far more could be made to contextualise this in the literature review. For example, by critically analysing a greater diversity of previous work on DEMs of Difference (P1L25) a stronger case could be made for the need to scale-up the typically small-scale topographic surveys that are acquired using terrestrial / airborne geomatics techniques to generate DEMs.

2) Context, methods and results need to be appropriate separated. For example, P2L20-28 is primarily methodological detail but in the introduction section. Much of the material on P4 is context for the research question (introductory material). Some of the material in section 3 is discussing methods or presenting results but this section comes before section 4 (methods).

3) The description of how "trunk channels" (P8L22) were digitised is confusing. Within the braided rivers literature, the term "trunk channels" is not widely used. Do you mean primary anabranch or the active width (i.e. Peter Ashmore's term)? This explanation (section 4.2.1) of the methods used to detect channel change needs to be improved (see also comments listed below). Fundamentally, it is not clear why a Level of Detection (LoD) approach for DEM differencing, rather than the now more widely used approach of probabilistic thresholding (see article by Wheaton 2010 that is cited in the manuscript). At the very least a clear justification of why a LoD approach was applied is needed. However, a stronger analysis could be presented if the DEMs of Difference were regenerated using a probabilistic approach.

4) A stronger geomorphic interpretation of the results (e.g. P19L1) could be achieved if there was a clearer geomorphic hypothesis to underpin the research at the outset. P19L21 mentions "field work" undertaken over the last decade. Is there supplementary field data that could be used to evaluate the remote sensing results from a more quantitative perspective?

5) The conclusion argues that "previous" measurements are constrained by high signal to noise ratios to detect vertical change. However, the noise magnitude reported from the satellite radar approach is significant. In my opinion contemporary approaches to DEM differencing are all challenged by difficulties separating geomorphic signals from noise when the vertical magnitude of change is relatively small compared to the elevation variations typically associated with particular geomorphic units that are under investigation. The conclusion would also benefit from a clearer summary of the actual method presented; the statement on P21L19 require more context within this section.

Minor comments

P1L3. The first sentence is focused on the cryosphere yet the paper is primarily focused with changes in terrain (rock / sediment). A more appropriate initial sentence is required.

P1L25. A greater diversity of refs is required for the rivers and earthquake examples.

P8L22. I think "hand picked" should say "digitised"

P8L29. Was there no vegetation at all? This is context dependent for gravel-bed rivers.

P8L24. "Error factors" need to be explained.

P19L23. A comment is required about the 0.2m/yr average rate to state that this assumes geomorphic work is constant each year.

P20L27. A clearer explanation of how field / auxiliary data could be used is needed.

---

## Author Comment (AC1) · 30 Aug 2018

Responses to Anonymous Referee #1 for manuscript (esurf-2018-51) submission to Earth Surface Dynamics:

**Measuring Decadal Vertical Land-level Changes from SRTM-C (2000) and TanDEM-X (~2015) in the South-Central Andes**

We appreciate the review and the improvements suggested by close reading of the manuscript. Highlighted in **bold are the reviewer comments** followed by our reply. All changes will be made to the final manuscript submission following completion of the interactive review period.

**Response to Anonymous Referee #1**

**The study analyses elevation differences between the SRTM and TanDEM-X DEMs over mountain terrain. The processing procedure to reach maximum accuracy and to remove various horizontal and vertical shifts is sound, but the geophysical results are more sparse than expected. My comments:**

**General comments**

**(1) The study is mostly a method presentation, the results are less spectacular than the reader expects ("novel", "first time"). I strongly recommend to tone done the latter type of announcements to make the reader not expect dense measurements over an entire mountain region.**

The manuscript primarily focuses on the methods (particularly the SRTM-C correction steps). The river and landslide measurements are used as examples of the method in action. We agree that the results in both cases are sparse and we have made an effort to tone down the language to reflect this. An excerpt of a changed sentence from P1L5-7 in the Abstract:

*Before:* For the first time we measure land-level changes at the scale of entire mountain belts in the south-central Andes using the SRTM-C (collected in 2000) and the TanDEM-X (collected from 2010–2015), both spaceborne radar DEMs.

*After:* Following careful corrections, we are able to measure land-level changes in gravel-bed channels and steep hillslopes in the south-central Andes using the SRTM-C (collected in 2000) and the TanDEM-X (collected from 2010-2015) near-global DEMs

**(2) Perhaps show and analyse (sediment fluxes) the entire river reach covered not only a subsection.**

We appreciate this comment and point out that this was our primary (initial) motivation: to establish a full-catchment vertical land-level dataset. However, during processing we realized the constraints and considered it more useful to elaborate on the methodological processing steps than a sediment budget. Specifically, our analysis is limited by SRTM-C resolution and channel width in the region. The raw signal of the SRTM-C has approximately 30 m ground resolution, however, this degrades to 45-60 m following post-processing in delivered products (see P4L18-20). To avoid the inclusion of non-channel regions (e.g., steep hillslopes) we apply a negative 60 m buffer to the bank-to-bank digitized channel polygon (see P8L22-25). Thus, only center pixels where the local channel width exceeds 120 m are considered in *dh* mapping. Channel widths further upstream of the analyzed reaches, and from their tributaries in the steep catchments that characterize this mountain front, have only sparse pixels meeting this requirement and are thus excluded.

**(3) Page5Line5, and else: land level changes are measured for instance using medium resolution ASTER data. Tome down "for the first time" if it refers to medium resolution. Also, are you sure that no landslides etc. have been measured using SRTM and TanDEM-X before? Also, TanDEM-X is in my view a high-resolution sensor, not a medium resolution one.**

As noted previously, we have toned down the language throughout the manuscript. Low, medium, and high-resolution terminology is relative and will change from decade to decade. In Passalacqua et al. (2015), the authors define high-resolution topography from sources like lidar and very high-resolution satellites (e.g., Pleiades and WorldView) as meter to sub-meter resolution. With a raw radiometric ground resolution of ~3.3 m, this places the TanDEM-X data outside of this realm. However, these terms are always relative and 10 years ago the 90 m SRTM-C data was considered "high-resolution". In general, we dislike these relative terms and they certainly add confusion, particularly for future readers. To avoid this, we have carefully gone through the manuscript and removed references to coarse, medium, and high-resolution. Instead we spell out the resolution,

referring to Pleiades and WorldView as "sub-meter resolution satellites" and only using terms like "coarser" or "finer" in relative references between datasets. e.g.:

P2L13: "Despite recent advances in meter to sub-meter lidar, satellite, and unmanned aerial vehicle data availability (Passalacqua et al., 2015), these remain limited in spatial and temporal coverage, and sometimes prohibitively expensive. Coarser gridded DEMs from radar and optical spaceborne sensors remain the best, and often only, option in large or remote areas."

Regarding landslides, we did not find any previous studies that specifically used the SRTM-C and newly released TanDEM-X data for mapping or volume estimation. However, recent work (e.g., Wessel et al., 2018) has begun examining the effect of land-cover on TanDEM-X, which could be useful in volume estimations of biomass and for assessing land-cover changes caused by deforestation, urbanization, and agriculture.

**(4) P2L6: be less strict. ASTER timeseries detect a few cm/dm per year over 10-20 years (=dm-m total change)**

We argue that ASTER requires either (a) many meters of elevation difference to overcome the large amount of noise in these low quality DEMs or (b) a long enough time series to identify trends in individual pixels. We therefore see no issue with the sentences (P2L4-9):

"Using DEMs from sources like the Advanced Spaceborne Thermal Emission and Reflection Radiometer (ASTER; Tachikawa et al. (2011)) with higher uncertainties is acceptable for monitoring glaciers and ice sheets (e.g., Brun et al., 2017), where $dh$ between even sub-annual time-steps can be tens to hundreds of meters over areas of many square kilometers. On the other hand, $dh$ of soil, rock, and unconsolidated sediment are often at the centimeter to meter scale and far more localized over up to a few hundred to thousand square meters."

**(5) P2L14: what about ArcticDEM, HighMountain Asia DEM, ALOS PRISM AW3D? I wouldn't call them limited spatial coverage.**

Arctic and High Mountain Asia DEMs are non-global datasets. ALOS PRISM is available at 5 m resolution for much of the globe, though major holes remain in high mountain regions. Furthermore, the quality of this data is suspect (please refer to our previous publication: Purinton

and Bookhagen, 2017) and does not match the quality expected from sub-meter sensors like WorldView or Pleiades. Furthermore, both the TanDEM-X and SRTM-C MEASURES DEM (the SRTM-C version used in this study) have been extensively referenced to ICESat measurements (see P6L8) and are referenced to the same geoid. Thus these datasets (in addition to lower quality ALOS AW3D and ASTER GDEM2 DEMs) represent unique DEMs, which may be widely applied by other scientists in diverse regions around the globe.

**(6) Section 3 (Correction...) is rather part of the methods.**
We agree with the suggestion and have moved this section to the beginning of the methods, thus removing Section 3.

**(7) Fig 8 and 9: you also need to show the results outside the masks in order to let the reader judge the statistical significance.**
In Figure 9 we are showing the full, un-masked *dh* map using statistical thresholding in slope-height-error-consistency bins (please refer to P9L7-12). Only *dh* values that are in the bottom and top 5% of pixels in their respective bins are mapped. For Figure 8 we are only considering the channel pixels. By restricting measurement to only the low-slope, vegetation-free, buffered channel pixels we ignore increased uncertainties from higher slopes and areas with dense vegetation and anthropogenic tampering on banks and nearby farmlands that would hide the significantly changing channel pixels within the statistical cutoffs. We argue that the lack of statistically significant pixels being mapped on the low-slope, vegetation-free, arid Altiplano-Puna Plateau region in Figure 9A is evidence that the method returns primarily true change pixels, while ignoring statistically insignificant changes. We hope that the points made in the discussion at P20L26-30 clearly point out the caveats of the method regarding this change detection:
"Given remaining noise in the datasets, change mapping is limited to large areas of coherent change (e.g., massive landslides) or specific low-slope, sparsely vegetated areas of interest such as wide gravel-bed rivers. In either case, field knowledge or auxiliary data (even in the form of GoogleEarth$^{TM}$) is necessary for accurate assessment of true change signals versus noise. In any case, the magnitude of change must be significantly above the expected uncertainty between

DEMs, which in the case of SRTM-C and TanDEM-X is as low as ~3 m on flat, partially vegetated terrain, and increasing with slope and topographic complexity."

**(8) Fig 8: I guess these magnitudes of changes would also be visible in ASTER time series, or ASTER versus SRTM (for instance;Castro et al. 2016 (Nature Comm Art 13585), Brun et al. 2017 (in your list), Girod et al. 2017 (doi: 10.3390/rs9070704), Wang et al. 2015 (doi: 10.3390/rs70810117))**

Actually we have experimented with ASTER DEMs in the area, but scene quality is very low and obscured in most cases by heavy cloud cover (this is an orographic barrier experiencing heavy precipitation in much of the downstream reaches at the mountain front). We were unable to find a good collection of ASTER scenes with which to assess the MICMAC methods of Girod et al. (2017) or regressions of Wang et al. (2015). Regarding the magnitudes of change, likely only the very large anthropogenic piles would be visible in an ASTER time series, given the > 5 m vertical uncertainty typically associated with even carefully hand-clicked ASTER DEMs in high relief areas, and worse still in lower quality ASTER DEMs (Purinton and Bookhagen, 2017). We added the following sentence at P4L8:

"Additionally, a dearth of cloud-free, high-quality ASTER imagery covering the study area precludes the automated DEM generation of Girod et al. (2017) and regression techniques of Wang et al. (2015)."

**(9) Fig 9: The landslide dh would also be visible without your processing, I guess. You could use this (and the river) to visualize the importance of your processing in more detail (before-after processing).**

Given the sparse pixels mapped, we have found that visual representation of the pre- and post-processed *dh* change maps are not so helpful. However, we think that this has been addressed textually in the case of the channels in the discussion at P19L6-10:

"Downstream of the knickpoint, Río Toro is in a net aggradation state with a corrected *dh* volume of $0.81\pm0.15\times10^6$ m$^3$, whereas, for Río Grande the net state is incision with a volume of -$0.69\pm0.15\times10^6$ m$^3$. In comparison, the pre-correction volume in each case is -$1.18\pm0.12\times10^6$ m$^3$

and 2.80±0.11×$10^6$ m$^3$ for Río Toro and Río Grande, respectively, thus indicating a flip in sign and reduction of magnitude following careful corrections applied prior to differencing."

And for the landslide we have added a clarification to the volume estimation at P20L8:

"These magnitudes of change show little difference in the pre- and post-corrected mapping, indicating (a) this is a localized region of good agreement between SRTM-C and TanDEM-X and (b) this large landslide can be identified in uncorrected difference maps."

**(10) P19L7-8: How are uncertainties computed? StDeviation, StError (if StError, how computed?). How aggregated and how voids filled?**

Error bars in Figure 7 are the RMSE taken from low-slope ($< 5°$) stable terrain. This is noted in the text and is also used as the level of detection cutoff following statistical outlier identification to further remove any suspect pixels well within expected TanDEM-X / SRTM-C noise. The uncertainties for volume estimation on P19L7-8 are clarified with the addition of this sentence in the methods at P9L5:

"Volume changes are calculated from the sum of pixel area (900 m$^2$) multiplied by vertical change with uncertainties taken as the level of detection RMSE and propagated via equation (15) in Lane et al. (2003)."

**(11) P30L31: it is not that easy to account for radar penetration! It exceeds often the actual elevation change signal, and the correction magnitudes applied here. See above ASTER studies, and others.**

Duly noted, we soften this section by changing it to the following:

"We posit that these correction steps may also be applied to cryospheric studies, however, radar penetration would need to be carefully considered first as this may exceed *dh* signals."

**Detail Comments**

**page 1, line 2: rewrite 1st sentence. "Vertical change is measured in the cryosphere...". I understand what you mean, but what is "measuring in the cryosphere"?**

*Before:* Vertical change is often measured in the cryosphere via digital elevation model (DEM) differencing to assess glacier and ice-sheet mass balances.

*After:* It is common to measure vertical changes of ice-sheets and glaciers in the arctic and high mountains via digital elevation model (DEM) differencing.

**P1L4: typically much smaller (landslides, as the one you show later, are a frequent exception with vertical changes on the same order of magnitude as glaciers).**

*Before:* On the ice-free earth, land-level change is much smaller in magnitude and thus requires more accurate DEMs for differencing and identification of change.

*After:* Excluding large landslides, on the ice-free earth land-level change is smaller in magnitude and thus requires more accurate DEMs for differencing and identification of change.

**P18L15: leaves stripes noise at the expense of preserving ... ? Other way round?**

*Before:* This leaves some stripe noise at the expense of preserving topographic signal.

*After:* This conservative approach retains the true topographic signal at the expense of remaining stripe noise.

Sincerely,
For both authors,

Ben Purinton
Universität Potsdam, Germany
[purinton@uni-potsdam.de](mailto:purinton@uni-potsdam.de)

**References**

Bagnardi, M., González, P. J., and Hooper, A.: High-resolution digital elevation model from tri-stereo Pleiades-1 satellite imagery for lava flow volume estimates at Fogo Volcano, Geophysical Research Letters, 43, 6267–6275, https://doi.org/10.1002/2016gl069457, 2016.

Brun, F., Berthier, E., Wagnon, P., Kääb, A., and Treichler, D.: A spatially resolved estimate of High Mountain Asia glacier mass balances from 2000 to 2016, Nature Geoscience, https://doi.org/10.1038/ngeo2999, 2017.

Girod, L.; Nuth, C.; Kääb, A.; McNabb, R. & Galland, O.: MMASTER: Improved ASTER DEMs for Elevation Change Monitoring, Remote Sensing, 704, 2017.

Lane, S. N., Westaway, R. M., and Murray Hicks, D.: Estimation of erosion and deposition volumes in a large, gravel-bed, braided river using synoptic remote sensing, Earth Surface Processes and Landforms, 28, 249–271, 2003.

Passalacqua, P., Belmont, P., Staley, D. M., Simley, J. D., Arrowsmith, J. R., Bode, C. A., Crosby, C., DeLong, S. B., Glenn, N. F., Kelly, S. A., Lague, D., Sangireddy, H., Schaffrath, K., Tarboton, D. G., Wasklewicz, T., and Wheaton, J. M.: Analyzing high resolution topography for advancing the understanding of mass and energy transfer through landscapes: A review, Earth-Science Reviews, 148, 174–193, https://doi.org/10.1016/j.earscirev.2015.05.012, 2015.

Purinton, B. and Bookhagen, B.: Validation of digital elevation models (DEMs) and comparison of geomorphic metrics on the southern Central Andean Plateau, Earth Surface Dynamics, 5, 211, https://doi.org/10.5194/esurf-5-211-2017, 2017.

Tachikawa, T., Kaku, M., Iwasaki, A., Gesch, D. B., Oimoen, M. J., Zhang, Z., Danielson, J. J., Krieger, T., Curtis, B., Haase, J., et al.: ASTER global digital elevation model version 2-summary of validation results, Tech. rep., NASA, 2011.

Wang, D. and Kääb, A.: Modeling glacier elevation change from DEM time series, Remote Sensing, 10117-10142, 2015.

Wessel, B., Huber, M., Wohlfart, C., Marschalk, U., Kosmann, D., and Roth, A.: Accuracy Assessment of the Global TanDEM-X Digital Elevation Model with GPS Data, ISPRS Journal of Photogrammetry and Remote Sensing, 139, 171–182, 2018.

---

## Author Comment (AC2) · 30 Aug 2018

Responses to Anonymous Referee #2 for manuscript (esurf-2018-51) submission to Earth Surface Dynamics:

**Measuring Decadal Vertical Land-level Changes from SRTM-C (2000) and TanDEM-X (~2015) in the South-Central Andes**

We appreciate the review and the improvements suggested by close reading of the manuscript. Highlighted in **bold are the reviewer comments** followed by our reply. All changes will be made to the final manuscript submission following completion of the interactive review period.

**Response to Anonymous Referee #2**

**My primary expertise with respect to this manuscirpt lies in the technique, and subsequent interpretation, of geomorphic change detection. At the outset, I would therefore like to emphasise that my review focuses upon the overall form of the manuscript and the technical component of the DEMs of Difference analysis. I do not have the technical expertise to scrutinise the detail of the remote sensing data processing; other reviewers should be sought for this elements. Overall, this manuscript presents an interesting and novel demonstration of how spaceborne radar DEMs can be used to detect vertical change in the Earth's surface. However, in my opinion, for this journal the manuscript needs to be reorganised to present a clearer research question/aim at the outset that is focused upon the geomorphological problem that is being investigated. There are also elements of the context, methods and results that are not organised in a classical research paper order. For the material that is presented, I do not see a reason why the context, methods and results can't be split into separate sections. I elaborate on these two items below, in addition to identifying further major and minor points.**

We acknowledge the reviewers' statement regarding their expertise and nonetheless appreciate a very thorough and helpful review of the manuscript regarding the geomorphic questions. We do, however, disagree that the paper needs significant reorganization to frame the study around a geomorphic question, since we intended the work as a primarily methods paper. As mentioned in the response to Referee #1 our initial motivation for the study was to establish a full-catchment

vertical land-level dataset. However, during processing we realized the constraints and considered it more useful to elaborate on the methodological processing steps than a sediment budget. Thus the focus has been shifted from the specific geomorphic question to a clear (we think) description (including some basic code) that allows other users to perform similar analysis in other terrain. We chose to submit the study to Earth Surface Dynamics given a focus on methods (e.g., Grieve et al., 2016a; Dietze, 2018) and data quality (e.g., Grieve et al., 2016b; Purinton and Bookhagen, 2017) in this journal. Therefore, we do not feel that any restructuring or major reframing is in order, but we have tried to accommodate the below suggestions.

**Major comments**

**1) A clear geomorphic research problem needs to be identified at the outset and backed up with appropriate context. P2L29 describes what will be included in the paper but there is a need for a more explicit geomorphic aim and associated set of objectives. The data processing methodology to generate a DEM of Difference is novel and far more could be made to contextualise this in the literature review. For example, by critically analysing a greater diversity of previous work on DEMs of Difference (P1L25) a stronger case could be made for the need to scale-up the typically small-scale topographic surveys that are acquired using terrestrial / airborne geomatics techniques to generate DEMs.**

We clarify the manuscript contents at P2L29 to read:

"In this submission we discuss the errors associated with each of these datasets and the corrections applied to mitigate uncertainties in their differencing for *dh* detection outside of the cryosphere. This is primarily a data quality and methods focused study. Geomorphic change detection is applied via correction and differencing of the TanDEM-X and SRTM-C over the south-central Andes in northwestern Argentina (Fig. 1) to identify and measure areas of *dh* in gravel-bed channels specifically and then across the landscape."

Regarding the literature review of *dh* studies, we feel that the paragraph on P2L3-15 clearly contextualizes the research with regards to previous studies of geomorphic change detection relying on small-scale or sparse datasets.

**2) Context, methods and results need to be appropriate separated. For example, P2L20-28 is primarily methodological detail but in the introduction section. Much of the material on P4 is context for the research question (introductory material). Some of the material in section 3 is discussing methods or presenting results but this section comes before section 4 (methods).**

As noted in the response to Referee #1, we have removed Section 3 and moved this to the beginning of Section 4. It is common in remote sensing studies to briefly introduce the datasets used early on, hence the brief description of the TanDEM-X and SRTM-C at P2L16-28, which follows well the contextualizing paragraph mentioned in the last reply at P2L3-15. By setting this up early on, we avoid any confusion about the datasets we are referring to, since a number of TanDEM-X and SRTM-C products exist and are often called the same thing despite different processing. Section 2 on P4 is important for contextualizing the correction technique and dataset details. We feel that the study is laid out well as is and clearly indicates why we are using the specific versions of SRTM-C and TanDEM-X data we mention in the introduction.

**3) The description of how "trunk channels" (P8L22) were digitised is confusing. Within the braided rivers literature, the term "trunk channels" is not widely used. Do you mean primary anabranch or the active width (i.e. Peter Ashmore's term)? This explanation (section 4.2.1) of the methods used to detect channel change needs to be improved (see also comments listed below). Fundamentally, it is not clear why a Level of Detection (LoD) approach for DEM differencing, rather than the now more widely used approach of probabilistic thresholding (see article by Wheaton 2010 that is cited in the manuscript). At the very least a clear justification of why a LoD approach was applied is needed. However, a stronger analysis could be presented if the DEMs of Difference were regenerated using a probabilistic approach.**

We change "trunk channel" on P8L22 to "active width of the primary channel branch (herein, trunk channel)". We apply statistical cutoffs and a final LoD given the coarser (30 m) nature of our data, as opposed to the fine (meter to sub-meter) lidar and SfM-MVS data used in other gravelbed river studies. We refer to this as a hybrid of the two techniques. The following is added at P8L18 to clarify the above concerns:

"Previous change mapping over gravel-bed channels has relied on level of detection cutoffs and probabilistic thresholding (e.g., Lane et al., 2003; Wheaton et al., 2010). These studies have, however, been developed for meter to sub-meter photogrammetric or lidar data. Here we use a hybrid approach of statistical outlier detection on the entire distribution of pixels followed by a level of detection cutoff for remaining pixels well within the bounds for expected noise between the datasets. Remaining uncertainties are primarily caused by speckle noise and terrain characteristics, with the biggest impact from slope."

**4) A stronger geomorphic interpretation of the results (e.g. P19L1) could be achieved if there was a clearer geomorphic hypothesis to underpin the research at the outset. P19L21 mentions "field work" undertaken over the last decade. Is there supplementary field data that could be used to evaluate the remote sensing results from a more quantitative perspective?**

As noted, this is chiefly a data quality and methods study, and we do not wish to over-emphasize the geomorphic implications. Rather we intend to point out some applications of the method and interesting results that can be attained. We do not have quantitative field data for the entire catchment or over the entire time-span (back to February 2000 when the SRTM-C was collected), which could aid in this analysis, and the statement is merely intended to emphasize that field observations (in addition to GoogleEarth historical imagery), support the growth of these gravel piles.

**5) The conclusion argues that "previous" measurements are constrained by high signal to noise ratios to detect vertical change. However, the noise magnitude reported from the satellite radar approach is significant. In my opinion contemporary approaches to DEM differencing are all challenged by difficulties separating geomorphic signals from noise when the vertical magnitude of change is relatively small compared to the elevation variations typically associated with particular geomorphic units that are under investigation. The**

**conclusion would also benefit from a clearer summary of the actual method presented; the statement on P21L19 require more context within this section.**

Previous measurements refer mostly to those using ASTER DEMs, which had much higher noise compared with TanDEM-X. In fact, there is very little noise in the TanDEM-X data, and it is emphasized in the study that the remaining signal to noise ratio in the measurements is primarily diluted by the lower quality SRTM-C data (see also our earlier publication: Purinton and Bookhagen, 2017). We modify P21L3-6 to read:

"Previous measurement of land-level changes at the scale of entire mountain belts has been restricted to the cryosphere, where the signal of snow and ice change outweighs the noise associated with DEMs used for differencing (typically ASTER or single TerraSAR-X / TanDEM-X CoSSC DEMs). On the other hand, studies outside of the cryosphere have relied on high-accuracy meter to sub-meter data at much smaller scales to measure height changes in rivers and hillslopes."

Furthermore, we add the following sentence to the end of the first paragraph of the conclusion:

"Noise from imperfect datasets continues to hinder signal detection in low magnitude geomorphic change detection, however, this study continues to push the envelope of the potential for change mapping using the data currently available to many scientists."

We feel that aside from these clarifications the conclusion is appropriately concise, without providing a detailed method summary aside from mentioning the correction of SRTM-C orbital biases followed by differencing and statistical avoidance of noise.

Additionally, we would like to point out that published DEM uncertainty values are typically an average over many land-cover types and terrain characteristics (slope, curvature, aspect). These values are usually not representative of individual situations, and we've tried to do this justice by separating change detection and analysis between steep and vegetated (hillslopes) and flat and vegetation-free (gravel-bed channels).

**Minor comments**

**P1L3. The first sentence is focused on the cryosphere yet the paper is primarily focused with changes in terrain (rock / sediment). A more appropriate initial sentence is required.**

We disagree and wish to contextualize the method in terms of the cryosphere, where most of the remote sensing change detection is carried out. Essentially we are presenting a snow and ice technique to the rock and sediment community.

**P1L25. A greater diversity of refs is required for the rivers and earthquake examples.**

The earthquake example is unique, but we have added to the rivers the citation for Lane et al. (2003), Wheaton et al. (2010), and Cook (2017).

**P8L22. I think "hand picked" should say "digitised"**

Changed.

**P8L29. Was there no vegetation at all? This is context dependent for gravel-bed rivers.**

The channels were vegetation free aside from a few sparse bushes of about 1-2 m height occurring in the furthest upstream reaches. This is a semi-arid environment and the climatic and hydrologic regime has been characterized in previous publications (Castino et al., 2017; Bookhagen and Strecker, 2012).

**P8L24. "Error factors" need to be explained.**

These are elucidated as the binning values on P8L31-P9L3, but a change is made to clarify this line earlier:

*Before:* Change mapping was done by separating the in-channel dh values into bins of contributing error factors and applying $5^{th}$ and $95^{th}$ percentile cutoffs to each bin, thus only taking the top (positive=aggradation) and bottom (negative=incision) 5% of outliers.

*After:* Change mapping was done by separating the in-channel dh values into bins of contributing error factors (local relief and TanDEM-X individual scene consistency) and applying $5^{th}$ and $95^{th}$ percentile cutoffs to each bin, thus only taking the top (positive=aggradation) and bottom (negative=incision) 5% of outliers.

**P19L23. A comment is required about the 0.2m/yr average rate to state that this assumes geomorphic work is constant each year.**

Added the sentence:

"This rate represents an average for the entire measurement period and assumes constant geomorphic change, whereas the true rates are more stochastic, following rainfall and anthropogenic activity variation."

**P20L27. A clearer explanation of how field / auxiliary data could be used is needed.**

*Before:* In either case, field knowledge or auxiliary data (even in the form of GoogleEarth™) is necessary for accurate assessment of true change signals versus noise.

*After:* In either case, field data (e.g., repeat total station or GPS surveys), field knowledge (e.g., via observations of incising reaches or roads damaged by aggrading channels), and/or auxiliary data (e.g., GoogleEarth™ historical imagery change mapping) are necessary for accurate assessment of the location of true change signals versus noise.

Sincerely,

For both authors,

Ben Purinton

Universität Potsdam, Germany

purinton@uni-potsdam.de

**References**

Bookhagen, B. and Strecker, M. R.: Orographic barriers, high-resolution TRMM rainfall, and relief variations along the eastern Andes, Geophysical Research Letters, 35, https://doi.org/10.1029/2007gl032011, 2008.

Castino, F., Bookhagen, B., and Strecker, M. R.: Oscillations and trends of river discharge in the southern Central Andes and linkages with climate variability, Journal of Hydrology, 555, 108–124, 2017.

Cook, K. L.: An evaluation of the effectiveness of low-cost UAVs and structure from motion for geomorphic change detection, Geomorphology, 278, 195–208, 2017.

Dietze, M.: The R package "eseis" – a software toolbox for environmental seismology, Earth Surface Dynamics, 6, 669-686, https://doi.org/10.5194/esurf-6-669-2018, 2018

Grieve, S. W. D., Mudd, S. M., Hurst, M. D., and Milodowski, D. T.: A nondimensional framework for exploring the relief structure of landscapes, Earth Surface Dynamics., 4, 309-325, https://doi.org/10.5194/esurf-4-309-2016, 2016a.

Grieve, S. W. D., Mudd, S. M., Milodowski, D. T., Clubb, F. J., and Furbish, D. J.: How does grid-resolution modulate the topographic expression of geomorphic processes?, Earth Surf. Dynam., 4, 627-653, https://doi.org/10.5194/esurf-4-627-2016, 2016b.

Lane, S. N., Westaway, R. M., and Murray Hicks, D.: Estimation of erosion and deposition volumes in a large, gravel-bed, braided river using synoptic remote sensing, Earth Surface Processes and Landforms, 28, 249–271, 2003.

Purinton, B. and Bookhagen, B.: Validation of digital elevation models (DEMs) and comparison of geomorphic metrics on the southern Central Andean Plateau, Earth Surface Dynamics, 5, 211, https://doi.org/10.5194/esurf-5-211-2017, 2017.

Wheaton, J. M., Brasington, J., Darby, S. E., and Sear, D. A.: Accounting for uncertainty in DEMs from repeat topographic surveys: improved sediment budgets, Earth Surface Processes and Landforms, 35, 136–156, 2010.